# FEATURE CONSISTENT 4D GAUSSIAN SPLATTING FOR REALISTIC DYNAMIC VIEW SYNTHESIS

## ABSTRACT

Dynamic novel view synthesis remains challenging due to the complexity of diverse motion patterns. In 4D Gaussians, the temporal dimension further complicates constraint formulation, making temporally consistent rendering difficult. To address this, we introduce 4D Feature Gaussian Splatting (F4DGS), a dynamic rendering algorithm that introduces feature consistency regularization to enable realistic rendering. This regularization jointly synchronizes hierarchical semantic features, velocity, and depth, ensuring coherent motion and appearance. We further extend the regularization beyond static alignment to capture temporal associations over continuous unit time intervals. F4DGS is the first rendering algorithm to explicitly couple velocity and depth for learning motion-consistent 4D representations, enabling high-fidelity, physically plausible rendering of dynamic content. Through comprehensive evaluations on monocular and multi-view dynamic datasets, F4DGS achieves real-time, high-resolution rendering and consistently outperforms existing methods across both quantitative and qualitative benchmarks. Notably, F4DGS achieves a 3.51 PSNR improvement on the Plenoptic dataset with comparable rendering speed and training time.

## 1 INTRODUCTION

3D reconstruction remains a core topic in computer vision, with novel view synthesis (NVS) playing a key role in applications such as immersive gaming, the film industry, and VR. Most methods target either static environments with time-invariant elements Mildenhall et al. (2020); Knapitsch et al. (2017); Hedman et al. (2018); Barron et al. (2022) or dynamic scenes with time-varying content Pumarola et al. (2021); Park et al. (2021b); Li et al. (2022b); Wu et al. (2020); Cheng et al. (2023). Dynamic scene rendering is fundamentally a high-dimensional optimization problem that demands not just increased computational power, but also innovative representations. Key directions include developing temporally-aware rendering algorithms that handle fine-grained appearance and designing physically grounded, spatiotemporally continuous frameworks that preserve visual fidelity despite motion discontinuities and occlusions.

To address dynamic novel view synthesis, many methods have been proposed to jointly model 3D geometry and scene dynamicsKanade et al. (1997); Zitnick et al. (2004); Li et al. (2012); Collet et al. (2015); Du et al. (2021); Gao et al. (2021). NeRF Mildenhall et al. (2020) achieves high-quality view synthesis by representing scenes as implicit functions and using volume rendering to link 2D images with 3D structures. However, its dense ray sampling results in high computational costs for both training and rendering. 3D Gaussian Splatting (3DGS) Kerbl et al. (2023) overcomes this by replacing volumetric rendering with efficient rasterization of explicit 3D Gaussians, enabling real-time rendering. Yet, modeling realistic appearance with Gaussians remains challenging and becomes even more pronounced in 4D Gaussian Splatting (4DGS) due to the added temporal dimension. The increased representational complexity makes it harder to enforce consistency across time, leading to temporal artifacts that degrade both appearance fidelity and semantic coherence in dynamic scene rendering.

Because of their smooth nature, Gaussian representations struggle to capture fine details in dynamic scenes with sparse inputs, resulting in blurred surfaces. The absence of explicit constraints linking Gaussians to photorealistic properties further leads to appearance inconsistencies, including lighting. These challenges are most pronounced in scenes with hierarchical textures, where edge

sharpness and texture alignment degrade noticeably. The transition from static 3D to dynamic 4D representations is increasingly complex, as maintaining temporal coherence among Gaussians becomes increasingly difficult. Temporal updates introduce instability, especially in regions with fine-grained textures, where appearance consistency is poorly constrained. While increasing Gaussian density can improve visual quality, it substantially raises computational and memory costs, posing a significant challenge for real-time and resource-limited applications.

To achieve consistent, realistic appearances in dynamic scenes, we propose feature consistency regularization, which combines 4D hierarchical OT-semantics regularization and 4D motion–depth regularization to explicitly model the temporal dimension in 4D space. We begin by extracting hierarchical semantic features that capture fine-grained textures and scene semantics. These features guide F4DGS in maintaining accurate scene understanding over time. For precise spatial alignment, we introduce 4D hierarchical OT-semantics regularization, which aligns the extracted semantic features with the rendered outputs. Unlike methods that naively apply OT distance Villani et al. (2008), our hierarchical OT regularization imposes multi-level constraints on the motion of Gaussian distributions corresponding to the same feature. This multi-scale semantic alignment ensures that Gaussians remain consistent with the underlying static scene structure, effectively preserving spatial coherence throughout motion.

However, only at individual time steps synchronizing multi-level semantic features, velocity, and depth is insufficient for modeling coherent motion in dynamic scenes. To address this, we extend feature consistency regularization to operate over unit time intervals, where temporal variations in motion features provide richer information than isolated timesteps. Our unit-time interval extension is especially crucial for 4D Gaussians, where maintaining consistency is inherently more challenging than in 3D due to the temporal dimension. By jointly constraining multi-level semantic features, velocity, and depth over unit time intervals, feature consistency regularization addresses inconsistencies in Gaussian representations of the same feature across time. F4DGS learns transferable, invariant motion features, effectively improving both the fidelity of appearance and the realism of motion over time. Moreover, temporal modeling drives pixel-level appearance changes that simulate complex light interactions, *e.g.*, reflection, and refraction, across curved surfaces, enhancing photorealism.

To enable physically realistic motion rendering, we further introduce 4D motion–depth regularization, the first approach to jointly integrate the dynamic velocity with time-varying depth modeling of Gaussian distributions. Velocity plays a critical role in maintaining temporal coherence, ensuring that motion appears smooth and physically plausible. By constraining velocity, our approach effectively suppresses unnatural behaviors, *e.g.*, abrupt directional shifts. Furthermore, we incorporate depth features to align Gaussian trajectories with the true motion paths and the semantic structure of the scene. This integration addresses occlusion artifacts and prevents visual inconsistencies, *e.g.*, unnatural jumps and distortions, especially during complex non-rigid interactions. 4D motion–depth regularization ensures that the evolution of Gaussian distributions follows real-world motion dynamics, enabling coherent, high-fidelity rendering in dynamic environments (shown in Figure 1). By integrating 4D hierarchical OT-semantics and motion-depth regularization, F4DGS addresses key challenges in dynamic scene rendering, excelling at rendering fine geometric details, non-rigid motion, and dynamic lighting effects in real-time. In summary, our contributions are as follows:

- The introduction of **feature consistency regularization**, effectively addressing the **motion inconsistency** caused by **complex motion patterns over time**.

- 4D motion-depth regularization substantially improving **the physical realism and temporal consistency** of dynamic scene rendering. To the best of our knowledge, F4DGS is the first rendering algorithm to precisely predict object motion by **integrating 4D Gaussian velocity with depth modeling over time**.

- 4D hierarchical OT-semantics regularization efficiently **integrating hierarchical semantic features into 4D Gaussians**, enabling **accurate scene understanding** for real-time rendering.

- An optimization approach of 4D Gaussian properties that dynamically refines Gaussian distributions by **synchronizing hierarchical semantic features, velocity, and depth**, effectively learning **motion-consistent representations** for dynamic scenes.

## 2 RELATED WORK

**NVS for Static Scenes.** To improve rendering quality, traditional methods rely on interpolation or geometric representations Levoy & Hanrahan (1996); Gortler et al. (1996); Buehler et al. (2001); Riegler & Koltun (2020), such as meshes Debevec et al. (1996); Thies et al. (2019); Waechter et al. (2014); Wood et al. (2023), voxel grids Kutulakos & Seitz (2000); Penner & Zhang (2017); Seitz & Dyer (1999), and multi-plane projections Flynn et al. (2019); Mildenhall et al. (2019); Srinivasan et al. (2019); Zhou et al. (2018). Neural Radiance Field (NeRF) Mildenhall et al. (2020) models static scenes through differentiable volume rendering Verbin et al. (2022); Müller et al. (2022); Fridovich-Keil et al. (2022), but real-time performance is hindered by the need to sample millions of rays Barron et al. (2021); Chen et al. (2022). While 3DGS Kerbl et al. (2023) enables real-time rendering, optimizing Gaussian distributions for realistic appearance remains a challenge. To address this, F4DGS integrates 4D hierarchical OT-semantics regu-

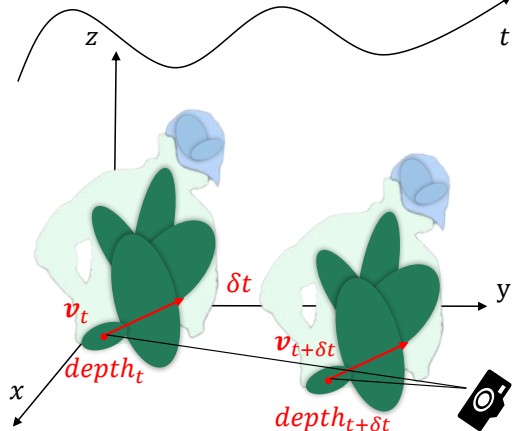

Figure 1: **A Simplified Illustration of the Feature Consistency Regularization.** Our approach jointly synchronizes hierarchical semantic features, velocity, and depth, extending beyond static alignment to model temporal associations over continuous time intervals.

larization, which aligns hierarchical semantic features to rendered features. This alignment effectively optimizes the underlying scene representation, enabling precise and photorealistic rendering of dynamic environments.

**NVS for Dynamic Scenes.** Generating novel views of dynamic scenes is challenging due to the need to model temporal dependencies. Different from existing rendering methods, which rely on geometric structures Li et al. (2012); Collet et al. (2015); Kanade et al. (1997); Zitnick et al. (2004), NeRFs Mildenhall et al. (2021); Li et al. (2022a;b); Fridovich-Keil et al. (2023); Cao & Johnson (2023) model motion either via canonical-space deformation fields Park et al. (2021a;b); Tretschk et al. (2021); Pumarola et al. (2021); Fang et al. (2022), flow-based constraints Du et al. (2021); Gao et al. (2021); Li et al. (2021); Guo et al. (2023); Tian et al. (2023), scene decomposition Song et al. (2023); Lee et al. (2024); Shao et al. (2023), and keyframe-driven dynamic representations Attal et al. (2023). However, NeRF-based methods suffer from high storage and computational costs Zhang et al. (2020); Wang et al. (2023); Gan et al. (2023). In contrast, 3DGS leverage GPU acceleration for real-time rendering Kerbl et al. (2023); Li et al. (2023); Yang et al. (2023b); Huang et al. (2024); Wu et al. (2023). Despite this, mainstream 3DGS-based methods lack effective temporal modeling, limiting their ability to render dynamic scenes consistently and realistically. To overcome this, we propose F4DGS, which integrates semantic understanding, motion prediction, and spatiotemporal regularization for physically realistic dynamic scene rendering.

## 3 METHODOLOGY

To achieve a realistic, physically-aware rendering, 4D Feature Gaussian Splatting (F4DGS) is composed of two key components: 4D hierarchical OT-semantics regularization and 4D motion-depth regularization (shown in Figure 2). To address visual inconsistencies across time, we introduce 4D hierarchical OT-semantics regularization to realize semantic understanding and temporal coherence in dynamic scenes (Section 3.1). We construct multi-level semantic features by combining CLIP's multi-modal embeddings with SAM's fine-grained visual representations, capturing both global context and local detail. These features are then aligned via multi-level optimal transport regularization, encouraging the rendered outputs to conform to the semantic structure of the real scene. To further address inconsistencies in the Gaussian distributions' trajectory that represent the same object or region across time, we extend the hierarchical OT-semantics formulation across unit time intervals, lifting it into the full 4D space–time domain. This temporal extension ensures that semantic features

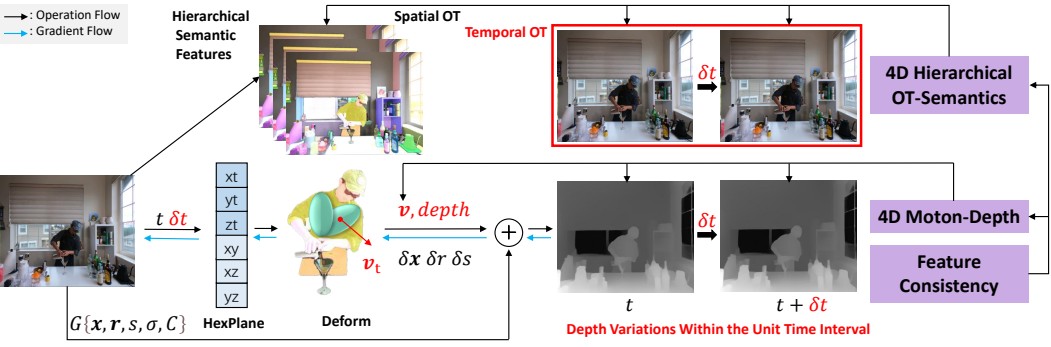

Figure 2: **Framework Overview.** F4DGS enables consistent, realistic rendering by jointly optimizing hierarchical semantic features, velocity $\delta v$, and depth $\delta depth$ over continuous unit time intervals $\delta t$. The resulting 4D Gaussians are differentiably rasterized into images and depth maps, while gradients from feature consistency regularization guide adaptive density control.

remain coherent across time, enforcing consistent object identity and spatial semantics throughout motion (Section 3.2). Furthermore, we introduce 4D motion–depth regularization, which couples velocity and depth to guide the motion of Gaussian distributions in a physically grounded manner. By leveraging these two complementary features as spatiotemporal priors, F4DGS gains the ability to predict motion trajectories accurately and address artifacts, especially in scenarios with sparse inputs and rapid dynamics (Section 3.3). Integrated with feature consistency regularization, which imposes both semantic and physical constraints, F4DGS learns transferable, high-fidelity 4D representations (Section 3.4). Our approach enables Gaussian distributions to evolve along smooth, physically plausible trajectories, including complex non-rigid motions, while maintaining real-time performance without increasing the point cloud size.

## 3.1 HIERARCHICAL OT-SEMANTICS REGULARIZATION

Rather than relying solely on direct optimal transport (OT) distances Villani et al. (2008), we introduce a hierarchical semantic regularization framework that leverages multi-scale semantic features to guide the motion of Gaussians over time. These features impose multi-level constraints on Gaussians representing the same object across time, enabling F4DGS to render the motion not just geometrically, but also semantically. Our semantic regularization steers F4DGS toward fine-grained, structure-aware rendering, ensuring that Gaussian trajectories remain not only visually realistic but also semantically and physically aligned with the underlying scene structure. As a result, F4DGS achieves robust rendering with improved consistency in both appearance and motion, including complex and occluded regions where direct photometric cues is unreliable.

We extract multimodal features at three semantic levels, i.e., coarse, middle, and fine, from both the input and the rendered outputs. Denote the hierarchical semantic features as $\{\mathbf{z}_i^{(\ell)}\}_{i=1}^{M_\ell} \subset \mathbb{R}^d$ and the corresponding rendered features as $\{\mathbf{h}_j^{(\ell)}\}_{j=1}^{N_\ell} \subset \mathbb{R}^d$, where $\ell \in \{\text{coarse}, \text{mid}, \text{fine}\}$. For each semantic level $\ell$, we define uniform empirical measures over the feature distributions:

$$P^{(\ell)} = \frac{1}{M_\ell} \sum_{i=1}^{M_\ell} \delta_{\mathbf{z}_i^{(\ell)}}, \qquad Q^{(\ell)} = \frac{1}{N_\ell} \sum_{j=1}^{N_\ell} \delta_{\mathbf{h}_j^{(\ell)}}. \qquad (1)$$

We construct a cost matrix $\mathbf{C}^{(\ell)} \in \mathbb{R}^{M_\ell \times N_\ell}$ based on cosine distance:

$$C_{ij}^{(\ell)} = 1 - \cos\big(\mathbf{z}_i^{(\ell)}, \mathbf{h}_j^{(\ell)}\big) = 1 - \frac{\mathbf{z}_i^{(\ell)} \cdot \mathbf{h}_j^{(\ell)}}{\|\mathbf{z}_i^{(\ell)}\| \, \|\mathbf{h}_j^{(\ell)}\|}. \qquad (2)$$

To align the feature distributions, we solve the entropy-regularized optimal transport problem. The feasible transport plan $\mathbf{T}^{(\ell)} \in \mathbb{R}^{M_\ell \times N_\ell}$ satisfies the marginal constraints: $\mathbf{T}^{(\ell)} \mathbf{1}_{N_\ell} = \frac{1}{M_\ell} \mathbf{1}_{M_\ell}$,

$\left(\mathbf{T}^{(\ell)}\right)^T \mathbf{1}_{M_\ell} = \frac{1}{N_\ell} \mathbf{1}_{N_\ell}$ The Sinkhorn-regularized OT distance is defined as:

$$H(\mathbf{T}^{(\ell)}) = -\sum_{i,j} T_{ij}^{(\ell)} \log T_{ij}^{(\ell)}, \quad \mathcal{U}\left(P^{(\ell)}, Q^{(\ell)}\right) = \left\{ T^{(\ell)} \geq 0 \mid T^{(\ell)} \mathbf{1} = \frac{1}{M_\ell}\mathbf{1}, \ \mathbf{T}^{(\ell)})^T \mathbf{1} = \frac{1}{N_\ell}\mathbf{1} \right\},$$

$$d_{\mathrm{OT}}^{\lambda}\left(P^{(\ell)}, Q^{(\ell)}; C^{(\ell)}\right) = \min_{T^{(\ell)} \in \mathcal{U}\left(P^{(\ell)}, Q^{(\ell)}\right)} \sum_{i,j} T_{ij}^{(\ell)} C_{ij}^{(\ell)} - \lambda H\left(T^{(\ell)}\right). \tag{3}$$

where $\lambda$ is the regularization strength. The final hierarchical OT-semantics loss is:

$$\mathcal{L}_{\mathrm{HierOT}} = \sum_{\ell \in \{\mathrm{coarse,mid,fine}\}} \alpha_\ell \, d_{\mathrm{OT}}^{\lambda}(P^{(\ell)}, Q^{(\ell)}; \mathbf{C}^{(\ell)}), \quad \text{with} \quad \sum_\ell \alpha_\ell = 1. \tag{4}$$

where the weights $\alpha_\ell$ are learnable and optimized end-to-end. Our hierarchical OT-semantics regularization ensures spatiotemporal consistency and semantic coherence in dynamic scenes by guiding the motion of Gaussian distributions over time. It leverages hierarchical semantic features to steer the rendering process, ensuring that the motion of the Gaussian distributions aligns with both geometric laws and physical structures.

### 3.2 4D TEMPORAL MODELING

Constraining hierarchical semantic features at separate time steps is insufficient for dynamic scene modeling, as it neglects temporal coherence critical for realistic rendering. To address this, we extend our hierarchical semantic constraints across unit time intervals, enabling F4DGS to capture temporal variations and maintain appearance consistency over time. This is especially important in dynamic rendering, where temporal transitions must be smooth for coherent motion synthesis.

A central challenge lies in preserving consistency for Gaussians representing the same feature across time, particularly during motion and occlusion transitions. Our proposed 4D hierarchical OT-semantics regularization imposes spatiotemporal constraints on semantic features, addressing abrupt semantic shifts and promoting temporally smooth dynamics. This mechanism proves especially beneficial in motion-blurred and sparsely observed regions, where semantic continuity reduces underfitting and preserves geometric stability without requiring denser point clouds. We quantify temporal semantic drift via the OT distance at each semantic level $\ell$ over the unit time interval $\delta t$, defined as:

$$\delta d_t^{(\ell)} = d_{\mathrm{OT}}^{\lambda}\left(P^{(\ell,t+\delta t)}, Q^{(\ell,t+\delta t)}; C^{(\ell,t+\delta t)}\right) - d_{\mathrm{OT}}^{\lambda}\left(P^{(\ell,t)}, Q^{(\ell,t)}; C^{(\ell,t)}\right). \tag{5}$$

We penalize these temporal deviations to encourage stable semantic evolution:

$$\mathcal{L}_{\mathrm{TemporalOT}} = \sum_{\delta t} \sum_{\ell \in D} \left\| \alpha_\ell \, \delta d_t^{(\ell)} \right\|_2^2, \quad D = \{\mathrm{coarse, mid, fine}\}. \tag{6}$$

This temporal OT loss complements our spatial OT alignment, promoting semantic consistency across time without increasing the number of Gaussians. The final 4D hierarchical OT-semantics loss integrates both spatial and temporal components:

$$\mathcal{L}_{\mathrm{4D\ hierarchical\ OT\text{-}semantics}} = \sum_{\delta t} \mathcal{L}_{\mathrm{HierOT}}(t) + \lambda_1 \mathcal{L}_{\mathrm{TemporalOT}}. \tag{7}$$

where the first term ensures semantic alignment at each time step, and the second term regularizes semantic transitions over time. The weighting parameter $\lambda_1$ balances static alignment and dynamic smoothness, leading to temporally coherent and geometrically consistent dynamic scene rendering.

### 3.3 4D MOTION-DEPTH REGULARIZATION

To ensure physically consistent rendering in dynamic scenes, we introduce 4D motion-depth regularization, which augments each Gaussian with velocity and depth attributes. These jointly enforce structural and temporal coherence in F4DGS. Velocity encodes both inter-object dynamics and local deformations, enabling F4DGS to model realistic motion trajectories and anticipate object positions across time. This predictive capability reduces redundant real-time computations and ensures smooth temporal transitions.

By applying motion constraints over unit time intervals, velocity acts as a temporal prior, discouraging nonphysical artifacts, *e.g.*, sudden direction reversals and trajectory discontinuities. Meanwhile, depth provides complementary geometric cues for surface structure inference and temporal consistency. Together, these features enable accurate motion prediction and effectively resolve spatiotemporal inconsistencies in fast and under-observed regions.

Recognizing the strong correlation between motion and geometry, we explicitly synchronize velocity and depth at each Gaussian primitive over time to preserve motion consistency. Let $d_i(t)$ denote the depth of the $i$-th Gaussian primitive at time $t$, and $\mathbf{v}_i(t)$ the 3D velocity vector. The motion-induced depth change is expressed as:

$$\dot{d}_i(t) = \frac{d_i(t + \delta t) - d_i(t)}{\delta t}. \tag{8}$$

We penalize discrepancies between this depth change and the predicted velocity, both in the z-direction and full 3D norm:

$$\mathcal{L}_{\text{4D motion-depth}} = \sum_{\delta t} \sum_{i=1}^{N} \left[ \alpha \big( \dot{d}_i(t) - v_{z,i}(t) \big)^2 + (1 - \alpha) \big( \dot{d}_i(t) - \|\mathbf{v}_i(t)\| \big)^2 \right]. \tag{9}$$

where $N$ is the total number of Gaussians and $\alpha \in [0, 1]$ balances the alignment. This regularization enhances F4DGS's ability to track complex motion patterns, including non-rigid and fluid, while maintaining temporal coherence. By leveraging depth as an auxiliary supervisory feature, it improves robustness in challenging conditions, not only in fast motion and sparse visibility, but also in motion blur. Furthermore, temporal coupling of motion and depth contributes to accurate surface rendering, enabling the synthesis of fine-grained textures and photorealistic light effects, including specular reflections and refractions, on dynamic objects. Overall, our spatiotemporal regularization effectively improves the fidelity, smoothness, and realism of dynamic scene rendering.

### 3.4 Feature Consistency

To ensure spatiotemporal semantic coherence in dynamic scene rendering, we define the total feature consistency regularization that integrates both semantic and motion supervision:

$$\mathcal{L}_{\text{4DFeature}} = \mathcal{L}_{\text{4D hierarchical OT-semantics}} + \lambda_2 \, \mathcal{L}_{\text{4D motion-depth}}. \tag{10}$$

where $\lambda_2$ controls the trade-off between semantic alignment and motion consistency. Building on the original 3D Gaussian Splatting (3DGS) framework Kerbl et al. (2023), we additionally incorporate photometric losses, namely the $\mathcal{L}_1$ pixel-wise difference and the Structural Similarity Index (SSIM), to compare rendered images against ground truth. The final objective for F4DGS is thus formulated as:

$$\mathcal{L}_{\text{F4DGS}} = \mathcal{L}_{\text{4DFeature}} + \lambda_3 \, \mathcal{L}_{\text{SSIM}} + \lambda_4 \, \mathcal{L}_{\text{tv}} + \lambda_5 \, \mathcal{L}_1. \tag{11}$$

where $\mathcal{L}_{\text{tv}}$ is the grid-based loss Wu et al. (2023) and $\lambda_{\mathcal{O}}$ is the learned hyperparameter. This composite loss encourages F4DGS to synthesize high-fidelity renderings while maintaining semantic coherence and physically plausible motion across time.

## 4 Experiments

### 4.1 Datasets and Implementation Details

We validate our approach on two established benchmarks that each introduce their own complexities in dynamic scene rendering. The **Plenoptic Video Dataset Li et al. (2022b)** features six real-world scenes, offering 20 viewpoints for training and a single, central view held out for evaluation; all images measure $1352 \times 1014$ pixels Li et al. (2022b). The **D-NeRF Dataset Pumarola et al. (2021)** comprises monocular video captures from eight separate scenes, with each scene providing 50–200 training frames, 10–20 validation frames, and 20 test frames, all uniformly resized to $800 \times 800$ pixels Pumarola et al. (2021). Our implementation leverages PyTorch Paszke (2019) running on a single NVIDIA RTX 3090 GPU, and we adopt the same optimizer settings as those used in 3DGS Paszke (2019).

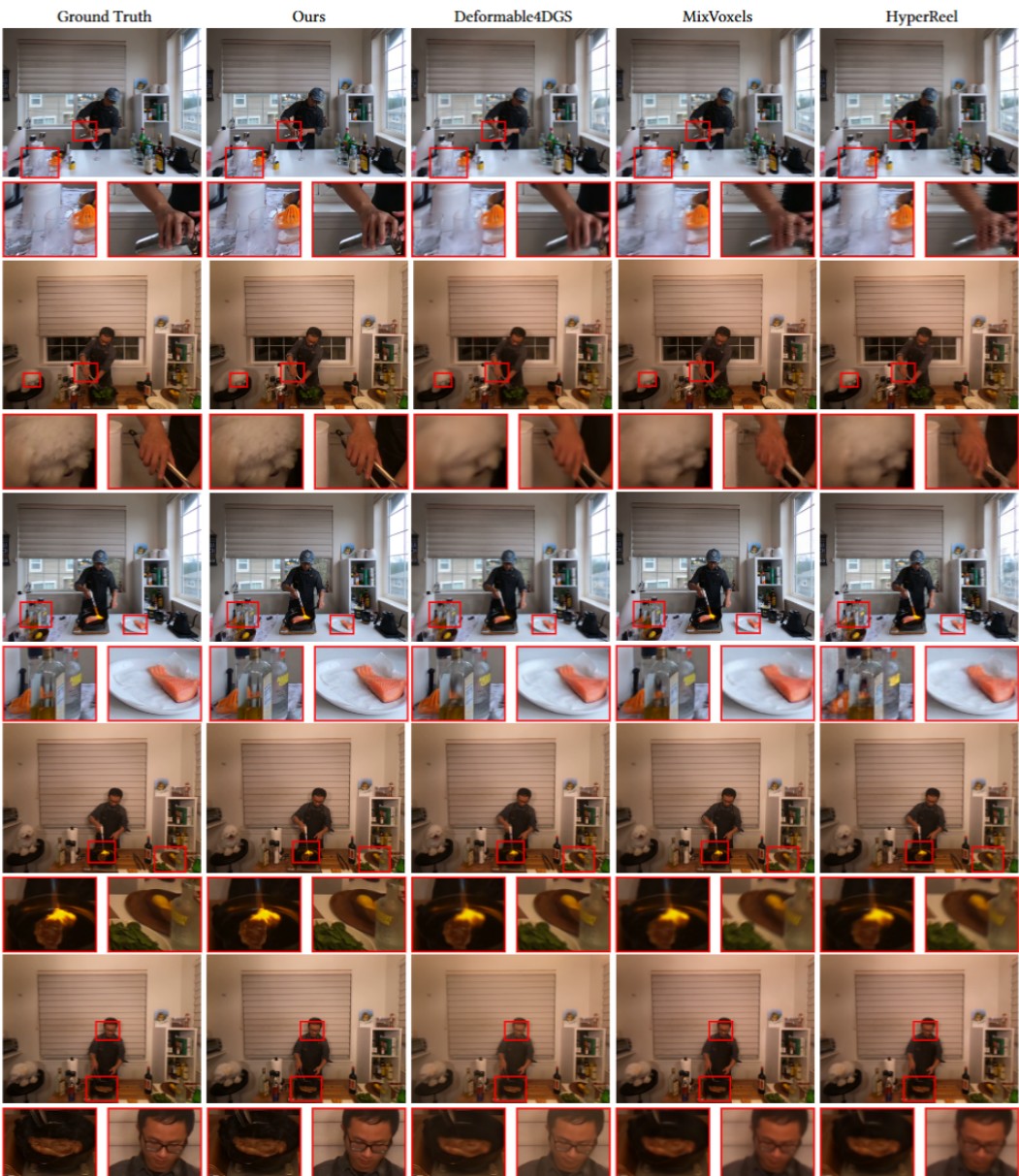

Figure 3: **Qualitative Comparison on Plenoptic Video Dataset**. From top to bottom, the visualized dynamic scenes are *Coffee Martini*, *Cook Spinach*, *Flame Salmon*, *Flame Steak*, and *Sear Steak*. F4DGS precisely renders fine-grained appearance of fast-moving objects and faithfully captures real-world physical behaviors, *e.g.*, the rapid motion of hands.

## 4.2 RESULTS

As shown in Table 1, NeRF-based methods face significant challenges in modeling four-dimensional dynamic scenes, largely due to the computational overhead of repeated neural network forward passes. Despite their intensive training requirements, these methods often fall short of delivering photorealistic renderings and are far from achieving real-time performance. Voxel-based approaches offer improved visual quality but still suffer from substantial training costs. In contrast, our F4DGS achieves notable improvements in both rendering speed and visual fidelity. It consistently outperforms existing methods in rendering high-resolution dynamic videos (1352×1014), delivering sharp details and temporal coherence. Compared to Deformable4DGS, F4DGS achieves a 3.51 PSNR

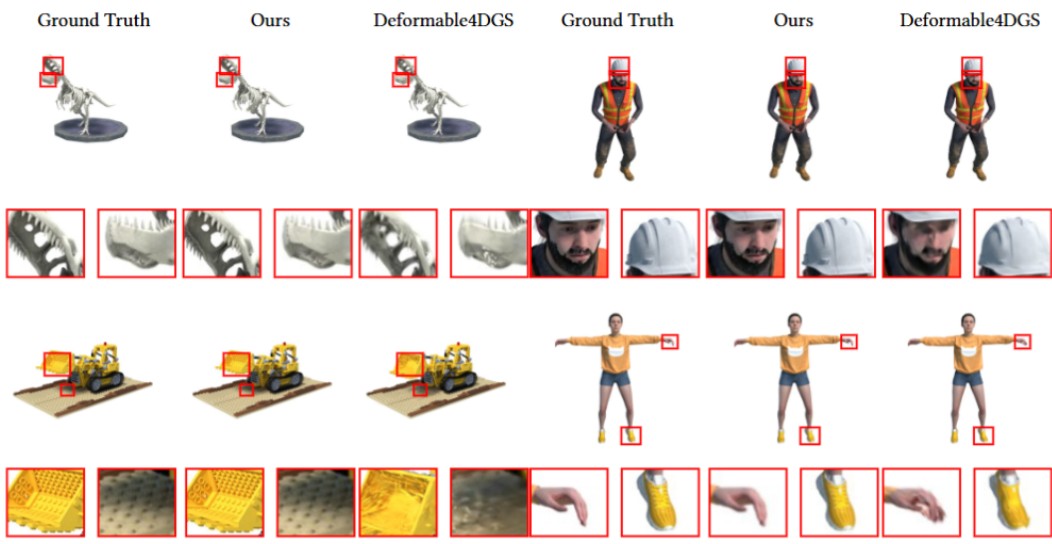

Figure 4: **Qualitative Comparison on D-NeRF Dataset.** From left to right, the visualized scenes include *T-Rex*, *Stand Up*, *Lego*, and *Jumpingjack*. We zoom in on high-frequency detail regions of fast-moving objects. F4DGS demonstrates precise rendering of multi-level surface textures under rapid motion, *e.g.*, the intricate facial expressions, and the complex shading distribution of the teeth.

gain, demonstrating a substantial leap in quality. While both methods utilize deformation–based geometric priors, the performance advantage of F4DGS stems from its feature consistency regularization, which facilitates the learning of motion-consistent 4D representations, ensuring both structural realism and efficient rendering.

As illustrated in Figure 3, F4DGS substantially improves visual fidelity over the baseline by rendering hierarchical fine-grained textures in dynamic scenes, *e.g.*, crisp lettering on transparent glass bottles (rows 3–4), salmon surface patterns at multiple granularities (row 3), and subtle color variations reflecting different degrees of steak doneness (row 5). Notably, F4DGS accurately renders complex reflections in motion, *e.g.*, the dynamic fluid-surface highlights within a glass (row 1). These improvements are driven by our feature consistency regularization, which resolves motion inconsistencies by the learning of motion-consistent, transferable representations that effectively capture nuanced and complex motion dynamics in real-world, reflective environments.

Table 1: **Quantitative Comparison on Plenoptic Video Dataset**. We compare F4DGS with both NeRF-based and Gaussian-based methods. F4DGS substantially outperforms the baselines in PSNR, achieving the shortest training time and comparative rendering speeds. *: trained on 8 GPUs and tested only on the Flame Salmon scene. The best , second best , and third best results are highlighted.

| ID | Method | PSNR↑ | SSIM↑ | LPIPS↓ | Train↓ | FPS↑ |
|---|---|---|---|---|---|---|
| a | DyNeRF Li et al. (2022b)* | 29.58 | - | 0.08 | 1344 h | 0.015 |
| b | StreamRF Li et al. (2022a) | 28.16 | 0.85 | 0.31 | 79 min | 8.50 |
| c | HyperReel Attal et al. (2023) | 30.36 | 0.92 | 0.17 | 9 h | 2.00 |
| d | NeRFPlayer Song et al. (2023) | 30.69 | - | 0.11 | 6 h | 0.05 |
| e | K-Planes Fridovich-Keil et al. (2023) | 30.73 | 0.93 | 0.07 | 190 min | 0.10 |
| f | MixVoxels Wang et al. (2023) | 30.85 | 0.96 | 0.21 | 91 min | 16.70 |
| h | RealTime4DGS Yang et al. (2023a) | 29.95 | 0.92 | 0.16 | 8 h | 72.80 |
| g | Deformable4DGS Wu et al. (2023) | 28.42 | 0.92 | 0.17 | 72 min | 39.93 |
| i | Ours | 32.50 | 0.96 | 0.07 | 25 min | 80.03 |

On the D-NeRF dataset, F4DGS achieves the highest rendering quality, improving the PSNR from 32.99 to 34.37 (Table 2). Figure 4 further highlights the advantages of F4DGS in capturing hierarchical semantic details, *e.g.*, the densely packed teeth (row 1) and the fine-grained textures (row 2). Under challenging reflective conditions, F4DGS also demonstrates precise rendering, exemplified by the precise shading distribution on the helmet (row 1). F4DGS effectively combines spatiotemporal consistency and semantic coherence, enabling high-precision, physically consistent rendering in complex dynamic scenes while maintaining efficient real-time performance. Through multi-level regularization strategies, it demonstrates exceptional robustness and rendering quality, even in the face of fast motion and sparse visibility.

### 4.3 ABLATION STUDIES

**4D Hierarchical OT-Semantics Regularization.**

To ensure accurate rendering of dynamic scenes, we introduce 4D hierarchical OT-semantics regularization, which aligns Gaussian distributions with the scene structure using OT distance. In isolation, this regularization enables F4DGS to learn richer semantics, *e.g.*, distinguishing the dog next to the human in Figure 5. This semantic guidance substantially enhances overall rendering quality, as shown in Table 3 and Figure 5. When combined with temporal modeling, F4DGS achieves its best performance, demonstrating the effectiveness of our temporal modeling approach.

Table 2: **Quantitative Comparison on D-NeRF Dataset**.

| Method | PSNR↑ | SSIM↑ | LPIPS↓ | Train↓ | FPS↑ |
|---|---|---|---|---|---|
| D-NeRF Pumarola et al. (2021) | 29.17 | 0.95 | 0.07 | 24 h | 0.13 |
| TiNeuVox Fang et al. (2022) | 32.87 | 0.97 | 0.04 | 28 min | 1.60 |
| K-Planes Fridovich-Keil et al. (2023) | 31.07 | 0.97 | 0.02 | 54 min | 1.20 |
| Deformable3DGS Yang et al. (2023b) | 39.31 | 0.99 | 0.01 | 26 min | 85.45 |
| RealTime4DGS Yang et al. (2023a) | 29.95 | 0.92 | 0.16 | 8 h | 72.80 |
| Deformable4DGS Wu et al. (2023) | 32.99 | 0.97 | 0.05 | 13 min | 104.00 |
| Ours | 37.80 | 0.98 | 0.01 | 6 min | 200.00 |

**4D Motion-Depth Regularization.**

To accurately capture the dynamic trajectory of moving objects, we propose 4D motion–depth regularization, the first approach to explicitly couple the 4D Gaussian distributions velocity with time-varying depth modeling. To assess its impact, we compare the base configuration with a variant that incorporates only this regularization. As shown in Table 3 and Figure 5, introducing accurate motion–depth constraints effectively enhances rendering quality, especially in tracking the full extent of object motion, *e.g.*, the movement range of the dog.

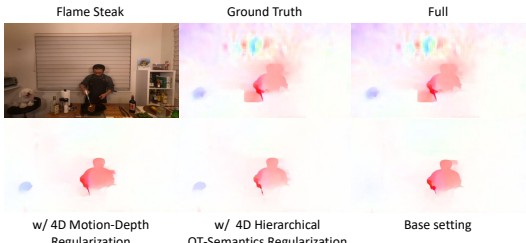

Figure 5: **Optical Flow Visualization.**

## 5 CONCLUSION AND FUTURE WORKS

We introduce 4D Feature Gaussian Splatting (F4DGS) for realistic dynamic novel view synthesis, addressing the challenges of temporal complexity and motion inconsistency in 4D Gaussian representations. Our feature consistency regularization synchronizes hierarchical semantic features, velocity, and depth, enabling temporally coherent rendering. Through unit time interval modeling, F4DGS improves motion consistency without increasing point cloud density. As the first method to couple velocity and depth for motion modeling, F4DGS enables physically-aware, high-fidelity rendering. It achieves the highest PSNR and consistently delivers realistic appearance across dynamic scenes, while maintaining competitive training time. Experimental results demonstrate F4DGS as a practical, scalable, and highly effective solution for photorealistic rendering in complex, dynamic environments.

Table 3: **Ablation Study with Quantitative Comparison on D-NeRF Dataset**. We validate feature consistency regularization on rendering quality PSNR: (*a*) Base setting, (*b*) OT (with 4D hierarchical OT-semantics regularization), (*c*) Motion (with 4D motion-depth regularization), and (*Full*) Time (with temporal consistency).

| ID | Ablation Items | | | D-NeRF | | |
|---|---|---|---|---|---|---|
| | OT | Motion | Time | Hook | Stand Up | Trex |
| *a* | | | | 30.99 | 35.12 | 31.74 |
| *b* | ✓ | | | 32.57 | 37.92 | 33.70 |
| *c* | | ✓ | | 32.77 | 38.28 | 33.98 |
| *Full* | ✓ | ✓ | ✓ | **33.12** | **38.31** | **34.53** |

**Limitations** There are two limitations. First, F4DGS may exhibit latency in adapting to sudden object or motion changes, primarily due to the absence of strong spatial priors. To address this, we plan to incorporate auxiliary signals that detect unexpected physical motion to improve responsiveness. Second, while F4DGS provides an efficient 4D representation, it still incurs high training costs on very large-scale scenes. Accurate reconstruction in such cases may require a divide-and-conquer training strategy.

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
