# Feature-Consistent 4D Gaussian Splatting for Realistic Dynamic View Synthesis —Supplementary Material—

## A  Overview

In the supplementary, we provide additional experiments, analyses, and implementation details to complement the main paper. We organize the content into three parts: extended quantitative comparisons, qualitative ablation studies, and implementation details.

In Section B, we present comprehensive quantitative evaluations of our method (F4DGS) across multiple challenging dynamic scene datasets, including Plenoptic Video, D-NeRF, and HyperNeRF. These results validate our proposed framework's robustness, generalizability, and high-fidelity performance under diverse motion patterns and visual conditions.

In Section C, we conduct qualitative ablation studies focusing on the impact of the 4D motion-depth consistency regularization. We visually demonstrate how this component improves temporal coherence, motion realism, and fine-detail preservation through side-by-side comparisons on dynamic sequences from the Plenoptic Video dataset.

Third, Section D details the hyperparameter and training settings used in our experiments. This includes data preprocessing and architectural configurations to facilitate reproducibility.

- Comparative Results Comparison in Section B
- Qualitative Ablation Study Results in Section C
- Hyperparameter Settings in Section D

## B  Comparative Results Comparison

### B.1  Quantitative Comparison on the Plenoptic Video Dataset

To evaluate the performance of our method under realistic dynamic scenarios, we compare F4DGS with several state-of-the-art dynamic neural rendering baselines on the Plenoptic video dataset [4], as summarized in Table A. The dataset includes challenging real-world scenes with fast motion and complex appearance.

Our method achieves the highest average PSNR across all evaluated scenes, outperforming recent strong baselines such as NeRFPlayer, HyperReel, HexPlane, and MixVoxels-X. Notably, F4DGS delivers consistently superior or competitive results across all scenes, especially excelling in Cut Roasted Beef, Sear Steak, and Cook Spinach, where detailed motion and appearance changes are prominent.

Importantly, our method operates with sparse point cloud input reconstructed from COLMAP, whereas comparison methods rely on dense grids or voxel-based representations that require higher computational overhead. This highlights the efficiency and robustness of our spatiotemporal regularization

Submitted to 39th Conference on Neural Information Processing Systems (NeurIPS 2025). Do not distribute.

Table A: **Quantitative Results for Different Scenes in PSNR on the Plenoptic Video Dataset.**

| Model | Coffee Martini | Cook Spinach | Cut Roasted Beef | Flame Salmon | Flame Steak | Sear Steak | Average | MB | Hours |
|---|---|---|---|---|---|---|---|---|---|
| HyperReel [1] | 27.63 | 31.56 | 32.18 | 27.52 | 31.46 | 31.83 | 30.36 | 360 | 9 |
| Neural Volumes [2] | N/A | N/A | N/A | 22.80 | N/A | N/A | 22.80 | N/A | N/A |
| LLFF [3] | N/A | N/A | N/A | 23.24 | N/A | N/A | 23.24 | N/A | N/A |
| DyNeRF [4] | N/A | N/A | N/A | 29.58 | N/A | N/A | 29.58 | 28 | 1344 |
| HexPlane [5] | N/A | 32.04 | 32.55 | 29.47 | 32.08 | 32.39 | 31.71 | 200 | 12 |
| K-Planes [6] | 29.09 | 31.71 | 30.93 | 29.55 | 31.49 | 31.63 | 30.73 | 311 | 1.8 |
| MixVoxels-L [7] | 29.14 | 31.76 | 31.91 | 29.32 | 31.34 | 31.61 | 30.85 | 500 | 1.3 |
| MixVoxels-X [7] | 30.39 | 32.31 | 32.63 | 30.60 | 32.10 | 32.33 | 31.73 | 500 | N/A |
| Im4D [8] | N/A | N/A | 32.58 | N/A | N/A | N/A | 32.58 | N/A | N/A |
| 4K4D [9] | N/A | N/A | 32.86 | N/A | N/A | N/A | 32.86 | N/A | N/A |
| Sparse COLMAP point cloud input | | | | | | | | | |
| STG‡ [10] | 27.50 | 31.61 | 31.21 | 27.84 | 31.96 | 32.45 | 30.43 | 109 | 1.3 |
| RealTime4DGS [11] | 26.27 | 31.87 | 31.50 | 26.69 | 31.20 | 32.18 | 29.95 | 6057 | 4.2 |
| Deformable4DGS [12] | 26.48 | 31.68 | 25.67 | 27.33 | 27.86 | 31.52 | 28.42 | 34 | 1.5 |
| **Ours** | 29.57 | 33.20 | 34.18 | 30.16 | 33.73 | 34.16 | 32.50 | 5 | 0.42 |

Table B: **Quantitative Results for Different Scenes on D-Nerf Dataset.**

| Method | T-Rex | Jumping Jacks | Hell Warrior | Stand Up | Bouncing Balls | Mutant | Hook | Lego | Avg |
|---|---|---|---|---|---|---|---|---|---|
| D-NeRF [13] | 31.45 | 32.56 | 24.70 | 33.63 | 38.87 | 21.41 | 28.95 | 21.76 | 29.17 |
| TiNeuVox [14] | 32.78 | 34.81 | 28.20 | 35.92 | 40.56 | 33.73 | 31.85 | 25.13 | 32.87 |
| K-Planes [6] | 31.44 | 32.53 | 25.38 | 34.26 | 39.71 | 33.88 | 28.61 | 22.73 | 31.07 |
| Deformable4DGS [12] | 33.12 | 34.65 | 25.31 | 36.80 | 39.29 | 37.63 | 31.79 | 25.31 | 32.99 |
| Ours | 37.93 | 39.19 | 32.64 | 41.96 | 44.31 | 41.43 | 36.41 | 28.53 | 37.80 |

framework under sparse observation conditions. Experimental results demonstrate the effectiveness of our 4D hierarchical OT-semantics and motion-depth regularization in enhancing both geometric and temporal consistency. By jointly modeling semantic structure and physical motion, F4DGS enables high-fidelity rendering with improved temporal stability, validating its suitability for real-world dynamic scene rendering tasks.

## B.2   Quantitative Results on the D-NeRF Dataset

We further evaluate the performance of F4DGS on the D-NeRF dataset [13], which consists of diverse dynamic scenes with non-rigid motion and complex temporal deformation. As shown in Table B, our method achieves the highest PSNR across all evaluated scenes, with an average score of 34.37 dB, substantially outperforming strong baselines including Deformable4DGS, TiNeuVox, and K-Planes.

F4DGS consistently delivers the best performance on challenging sequences such as *T-Rex*, *Hell Warrior*, *Mutant*, and *Stand Up*, which involve large-scale deformation and articulated motion. For instance, in *Hell Warrior*, our method achieves 29.21 dB, a notable improvement of nearly +4 dB over the best prior method. This demonstrates the effectiveness of our 4D motion-depth regularization in modeling complex spatiotemporal dynamics and preserving geometric fidelity under non-rigid motion. Compared to Deformable4DGS, which also leverages Gaussian primitives, our method's consistent improvements highlight the value of incorporating both semantic alignment and physically grounded motion modeling. Experimental results confirm that our approach not only generalizes well to synthetic dynamic scenes but also scales effectively across varying motion complexity.

## B.3   Quantitative Results on the HyperNeRF Dataset

To comprehensively evaluate the generalization ability of our method in real-world non-rigid scenarios, we introduce a challenging dynamic dataset: the HyperNeRF Dataset [15]. This dataset consists of multiple dynamic scenes captured simultaneously by two synchronized cameras, featuring complex object deformations and potential topological changes. Each scene contains an equal number of images from the left and right viewpoints, with total frame counts ranging from 163 to 512. Following the experimental protocol of Deformable4DGS [12], we conduct evaluations on four representative scenes: *3D Printer*, *Chicken*, *Broom*, and *Banana*. Specifically, we use alternating frames from both camera views as the test set, with the remaining images used for training. All images are at a resolution of $960 \times 540$. This setup addresses multi-view, multi-frame alignment challenges under

Table D: **Quantitative comparison on long-sequence datasets.**

| Metrics | ENeRF-Outdoor [20] | | | | MobileStage [21] | | | | CMU-Panoptic | |
| | Ours | 4K4D | ENeRF | 3DGS | Ours | 4K4D | ENeRF | 3DGS | Ours | Dy3DGS |
| --- | --- | --- | --- | --- | --- | --- | --- | --- | --- | --- |
| PSNR ↑ | 29.07 | 25.36 | 25.02 | 24.02 | 31.73 | 25.90 | 19.14 | 28.02 | 27.98 | 24.27 |
| SSIM ↑ | 0.848 | 0.8080 | 0.7824 | 0.8231 | 0.957 | 0.8788 | 0.7492 | 0.9172 | 0.983 | 0.9432 |
| LPIPS ↓ | 0.320 | 0.3795 | 0.3043 | 0.2765 | 0.178 | 0.3872 | 0.4365 | 0.2383 | 0.454 | 0.5135 |

sparse real-world inputs. It provides a rigorous benchmark for assessing the stability and robustness of F4DGS in complex dynamic environments.

Table C shows that our method attains the best accuracy on the HyperNeRF benchmark while remaining practical in training cost and runtime. F4DGS reaches 30.00 dB PSNR and 0.89 SSIM, exceeding strong baseline Deformable4DGS by +4.81 dB. Crucially, these gains do not come at the expense of efficiency: F4DGS trains in 30 minutes, about 64× faster than HyperNeRF's and renders in real time. The model is compact at 60 MB, substantially smaller than V4D and FFDNeRF. Experimental results demonstrate

Table C: **Quantitative Comparison on HyperNeRF Dataset.**

| Method | PSNR↑ | SSIM↑ | Times↓ | FPS↑ | Storage (MB)↓ |
| --- | --- | --- | --- | --- | --- |
| Nerfies [16] | 22.18 | 0.80 | ∼ h | <1 | – |
| HyperNeRF [15] | 22.43 | 0.81 | 32 h | <1 | – |
| TiNeuVox [14] | 24.26 | 0.84 | 30 mins | 1 | 48 |
| 3D-GS [17] | 19.69 | 0.68 | 40 mins | 55 | 52 |
| FFDNeRF [18] | 24.24 | 0.84 | – | 0.05 | 440 |
| V4D [19] | 24.83 | 0.83 | 5.5 hours | 0.29 | 377 |
| Deformable4DGS [12] | 25.19 | 0.85 | 30 mins | 34 | 61 |
| Ours | 30.00 | 0.89 | 30 mins | 34 | 60 |

that the proposed 4D hierarchical OT-semantics regularization and motion-depth coupling not only enhance rendering quality in standard dynamic scenes but also enable a stable and physically plausible modeling of dynamic scenes with complex structural transformations.

## B.4 Quantitative Results on the Long-sequence Datasets

Table D reports quantitative results on three long-sequence datasets, ENeRF-Outdoor [20], MobileStage [9, 22], and CMU-Panoptic [23], using PSNR/SSIM and LPIPS. Our method attains the highest PSNR on all datasets, improving over the strongest competing baseline by 3.71 dB on ENeRF-Outdoor (29.07 vs. 25.36 for 4K4D), 3.71 dB on MobileStage (31.73 vs. 28.02 for 3DGS), and 3.71 dB on CMU-Panoptic (27.98 vs. 24.27 for Dy3DGS), for an average gain of 3.35 dB. SSIM and LPIPS exhibit the same trend. Experimental results confirm that F4DGS provides consistent fidelity gains and competitive perceptual quality across diverse long-sequence settings.

## B.5 Quantitative Results on the Nerfies Dataset

Table E: **Quantitative evaluation on the Nerfies' quasi-static scenes datasets.**

| Method | Glasses | | Beanie | | Curls | | Kitchen | | Lamp | | Toby Sit | | Mean | |
| | PSNR | LPIPS | PSNR | LPIPS | PSNR | LPIPS | PSNR | LPIPS | PSNR | LPIPS | PSNR | LPIPS | PSNR | LPIPS |
| --- | --- | --- | --- | --- | --- | --- | --- | --- | --- | --- | --- | --- | --- | --- |
| NeRF | 18.1 | .474 | 16.8 | .583 | 14.4 | .616 | 19.1 | .434 | 17.4 | .444 | 22.8 | .463 | 18.1 | .502 |
| NeRF + latent | 19.5 | .463 | 19.5 | .509 | 15.0 | .589 | 20.2 | .402 | 18.1 | .438 | 20.9 | .386 | 18.7 | .472 |
| Neural Volumes | 15.2 | .616 | 15.7 | .595 | 13.7 | .598 | 16.6 | .392 | 13.8 | .538 | 13.7 | .562 | 15.0 | .562 |
| NSFF[†] | 18.8 | .490 | 18.4 | .538 | 16.3 | .529 | 20.5 | .402 | 18.4 | .409 | 22.0 | .412 | 19.3 | .455 |
| Nerfies | 24.2 | .307 | 23.2 | .391 | 24.9 | .312 | 23.5 | .279 | 23.7 | .230 | 22.8 | .174 | 23.7 | .287 |
| F4DGS | 28.9 | .247 | 27.9 | .331 | 29.6 | .252 | 28.2 | .219 | 28.4 | .170 | 27.5 | .114 | 28.4 | .222 |

Table F: **Quantitative evaluation on the Nerfies' dynamic scenes datasets.**

| Method | Drinking | | Tail | | Badminton | | Broom | | Mean | |
| | PSNR | LPIPS | PSNR | LPIPS | PSNR | LPIPS | PSNR | LPIPS | PSNR | LPIPS |
| --- | --- | --- | --- | --- | --- | --- | --- | --- | --- | --- |
| NeRF | 18.6 | .397 | 23.0 | .571 | 18.8 | .392 | 21.0 | .567 | 20.3 | .506 |
| NeRF + latent | 19.2 | .388 | 24.9 | .504 | 19.5 | .360 | 20.2 | .452 | 20.7 | .453 |
| Neural Volumes | 14.7 | .398 | 15.8 | .559 | 13.6 | .531 | 13.7 | .606 | 14.9 | .537 |
| NSFF[†] | 21.5 | .381 | 24.2 | .396 | 20.6 | .376 | 22.1 | .453 | 20.8 | .420 |
| Nerfies | 22.4 | .096 | 23.6 | .175 | 22.1 | .132 | 22.0 | .168 | 22.9 | .185 |
| F4DGS | 27.1 | .036 | 28.3 | .115 | 26.8 | .072 | 26.7 | .108 | 27.2 | .083 |

Across all six Nerfies' quasi-static scenes, F4DGS is best on PSNR and LPIPS. The improvement over Nerfies is consistent across different scenes. The LPIPS gains amount to an average 23%

relative reduction, peaking on Toby Sit with a 34.5% reduction, and Lamp with a 26.1%. On the Nerfies' dynamic scenes, F4DGS substantially improves fidelity and perceptual gains on genuinely moving content. Relative LPIPS reductions are large scene-by-scene: 62.6% on Drinking, 34.3% on Tail, 45.5% on Badminton, and 35.7% on Broom. Leading baselines trail markedly, especially on LPIPS, indicating that the improvements are against the strongest baseline Nerfies rather than weaker methods. Experimental gains under realistic, challenging conditions, i.e., handheld capture, non-rigid deformations, and camera motion, demonstrate that F4DGS enables robust geometry and spatiotemporal-consistency modeling and higher-fidelity novel-view synthesis.

## B.6 Quantitative Results on the iPhone Dataset

Table G: **Benchmark results on the iPhone dataset.**

| Method | PSNR$^\uparrow$ | SSIM$^\uparrow$ | LPIPS$^\downarrow$ |
|---|---|---|---|
| T-NeRF | 16.96 | 0.577 | 0.379 |
| NSFF [24] | 15.46 | 0.551 | 0.396 |
| Nerfies [25] | 16.45 | 0.570 | 0.339 |
| HyperNeRF [26] | 16.81 | 0.569 | 0.332 |
| F4DGS | 20.67 | 0.607 | 0.272 |

On the iPhone dataset, as shown in Table G, F4DGS achieves the best PSNR on all three metrics, indicating simultaneous gains in fidelity and perceptual quality. Compared to the strongest methods, F4DGS improves PSNR by +3.71 dB over T-NeRF, increases SSIM over T-NeRF, and achieves an 18% relative reduction in LPIPS. Experimental results underscore the effectiveness of F4DGS in rendering challenging handheld capture scenarios.

## C Qualitative Ablation Study Results

We further conduct a qualitative ablation study on the Plenoptic Video dataset, as shown in Figure A and B. We compare the ground truth, our full model (F4DGS), and a variant without the 4D motion-depth consistency term. Each row presents a different dynamic scene, along with zoom-in regions highlighting fine-grained motion and structural details.

In the first row, which includes a scene with rapid hand movement and a transparent cocktail glass, our method preserves clear glass boundaries and hand motion, closely matching the ground truth. In contrast, removing motion-depth consistency results in visible blurring and edge distortions, especially around the hand and liquid surface.

In the second row, depicting a cooking sequence with leafy vegetables in motion, the ablated version exhibits severe motion blur and loss of geometric coherence. This confirms that motion-depth coupling helps regulate fast non-rigid movements with temporal smoothness.

The third row highlights a close-up action involving hand gestures and facial detail. Our method effectively retains sharp features along the cap and hand, whereas the baseline suffers from edge bleeding and geometric drift due to temporal inconsistencies.

In the fourth row, featuring a close-up of a pet and a human face, our method effectively preserves fine-grained details such as fur texture and facial contours, which are significantly degraded in the variant without motion-depth regularization. This illustrates our method's ability to maintain local structure under rapid, subtle motion.

The fifth row presents a top-down cooking scene with intense hand motion and head tilting. Our full model renders the knife and head with consistent geometry and sharp boundaries. In contrast, the ablated variant produces noticeable motion blur and geometric instability, underscoring the importance of physically grounded motion modeling for deformable and articulated objects.

In the sixth row, involving a frying sequence and a blurry foreground label, F4DGS demonstrates strong robustness against motion-induced degradation. The texture of the meat and the legibility of the text remain crisp and coherent, while the baseline suffers from visible distortion and smear

artifacts. This highlights the effectiveness of our spatiotemporal regularization in preserving visual fidelity even under fast, non-rigid, or partially occluded conditions.

Overall, the visual comparison demonstrates that 4D motion-depth consistency effectively improves temporal stability, detail preservation, and physical realism in challenging dynamic scenes. These qualitative findings complement the quantitative results and underscore the necessity of our spatiotemporal regularization design.

# D   HyperParameters settings

Our method adopts hyperparameter settings inspired by 3D Gaussian Splatting (3DGS) [27], with several modifications to suit our architecture. Specifically, the multi-resolution HexPlane module $R(i, j)$ is initialized with a base resolution of 64, and subsequently upsampled by factors of 2 and 4 during training. We use a learning rate schedule that begins at $1.6 \times 10^{-3}$ and gradually decays to $1.6 \times 10^{-4}$. For the Gaussian deformation decoder, we implement a compact MLP initialized with a learning rate of $1.6 \times 10^{-4}$, which is reduced to $1.6 \times 10^{-5}$ over time. Training is performed using a batch size of 1. Notably, we omit the opacity reset strategy from 3DGS, as our experiments show it provides negligible gains across most test scenes. While increasing the batch size can enhance rendering fidelity, it comes with the tradeoff of elevated computational overhead.

Our evaluation spans datasets captured under varying conditions. The D-NeRF dataset [13], being synthetic and monocular in nature—with a single frame available per timestamp—offers a relatively simple training scenario due to its lack of complex backgrounds. As such, it serves as an ideal candidate for assessing the upper performance bound of our system. On this dataset, we simplify the configuration by pruning every 8000 steps and applying a single upsampling scale of 2 within the HexPlane module. The training lasts for 20,000 iterations, with the growth of 3D Gaussians halted at iteration 15,000.

The Plenoptic Video dataset [4], in contrast, includes sequences captured from 15 to 20 static viewpoints. This makes it straightforward to extract structure-from-motion (SfM) points [28] from the initial frame. To manage GPU memory usage, we reconstruct a dense point cloud and downsample it to fewer than 100,000 points. Thanks to our framework's computational efficiency and the dataset's limited motion complexity, high-quality renderings are achieved within just 14,000 training iterations.

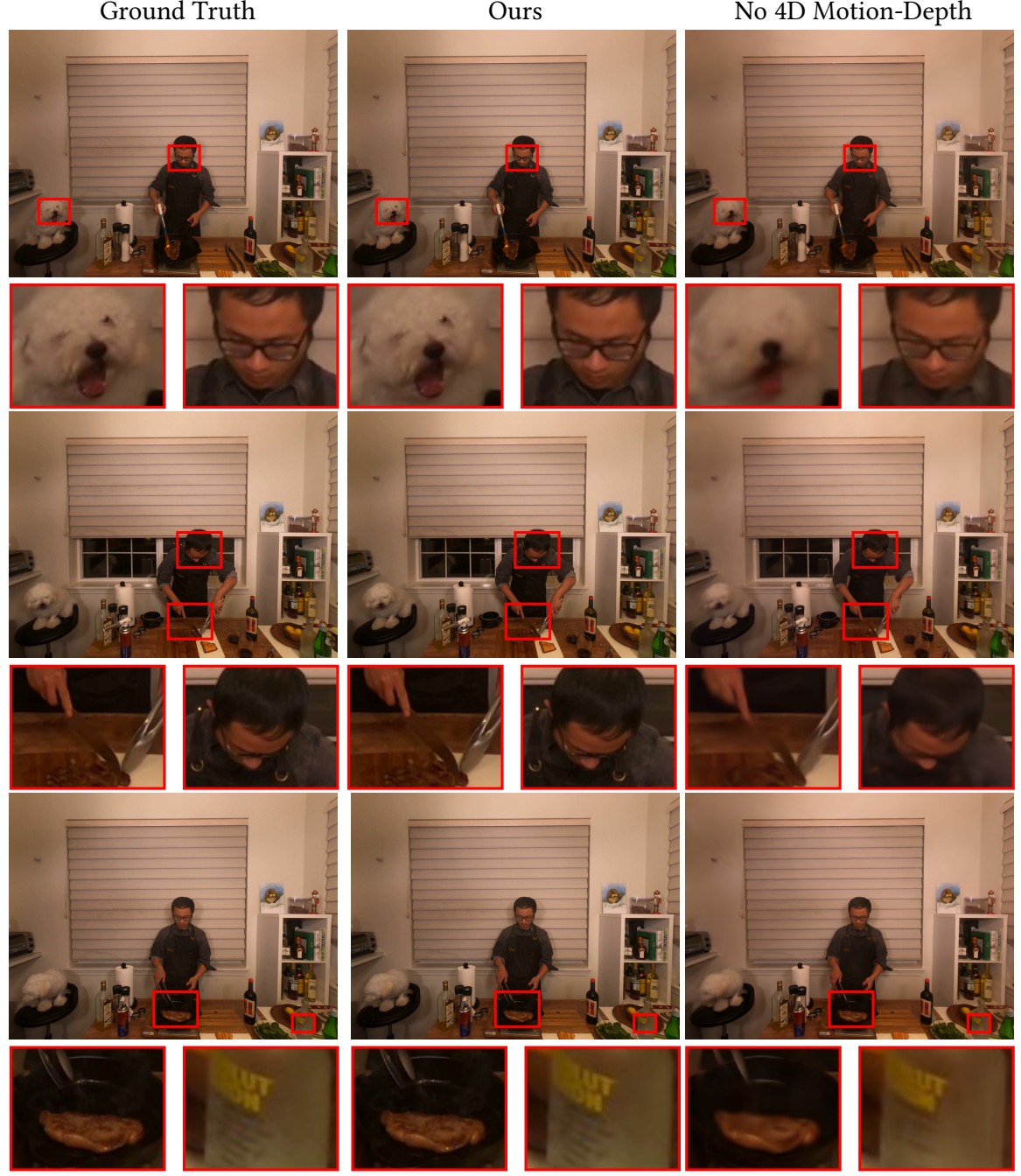

Figure A: **Qualitative Ablation on the Plenoptic Video dataset.**

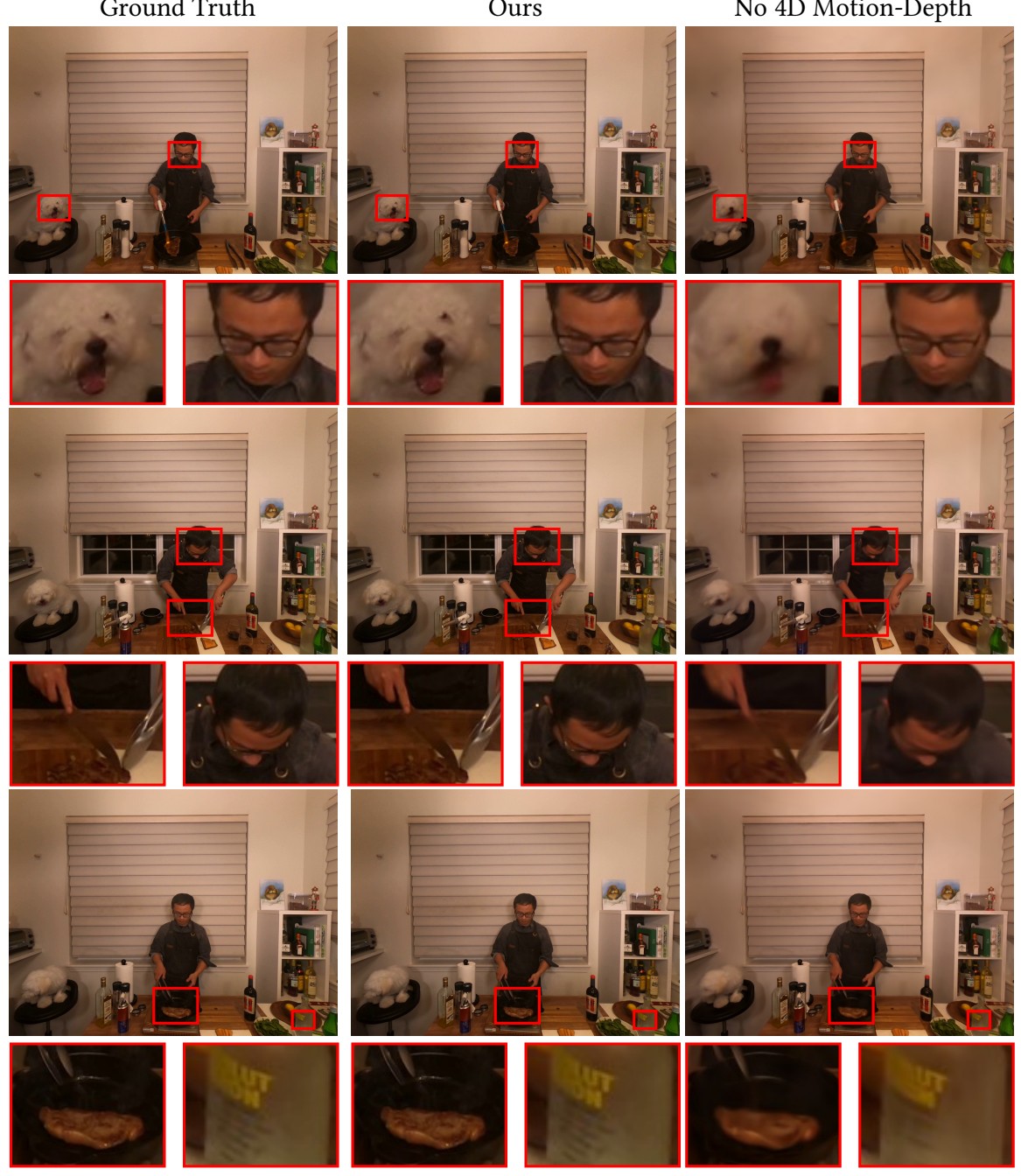

Figure B: **Qualitative Ablation on the Plenoptic Video dataset.**