# OpenReview forum: "Feature Consistent 4D Gaussian Splatting for Realistic Dynamic View Synthesis"
_ICLR.cc/2026/Conference — Submitted to ICLR 2026_

### Official Review · Reviewer_BhQR · 2025-10-27

**Soundness:** 1
**Presentation:** 1
**Contribution:** 2
**Rating:** 2
**Confidence:** 4

**Summary:**

This manuscript proposes a dynamic novel view synthesis framework, and claims that its integration of feature consistency regularization can facilitate more realistic rendering. The first part of this regularization aims to align the input and rendered multi-level semantic features by optimizing their optimal transport distance. And they try to improve the temporal consistency by constraining the OT distance to remain stable across frames. The second part augment each Gaussian with velocity and depth attributes and aims to encourage physically realistic motion by optimizing depth-flow consistency on them.

**Strengths:**

The idea of incorporating semantic consistency to regularize Gaussian Splatting’s optimization is interesting. Ideally, such semantic cues have the potential to help Gaussians preserve consistent object identities in regions where texture boundaries are ambiguous, by leveraging larger context and priors.

**Weaknesses:**

1.	The poor presentation makes this manuscript seems less serious. Many crucial details are missing in both the main text and appendix，which jeopardizes the reproducibility of some key components and justification of their design choice:
   -	The core contribution of this work seems to be the hierarchical OT–semantic regularization, yet the construction of hierarchical semantic features has not been clearly described. For instance, the ${z_{i}^{l}}$ in Line 202 is introduced without a definition or explanation of how it is exactly obtained.
   -	Please provide a clear definition of the unit time interval and specify the summation ranges in Equations (6) and (9).
   -	What does the depth and velocity attributes in L266 exactly refer to? Are they just some appended parameters like SH coefficients? If so, what guarantees that “velocity encodes both inter-object dynamics and local motion”? It seems that there is nothing to guarantee these attributes correspond to the literal velocity and depth of each Gaussian, and thus Equation (9) is not a reasonable objective, e.g., a trivial solution where $d$ is constant and $v=0$ could minimize it but is meaningless. Even if they indeed represent velocity and depth, the second term in Equation (9) appears to merely to force the motion along the depth dimension.
One the other hand, many statements sound empty or overclaimed:
   -	Why is the motion feature “transferable and invariant” (L078)?
   -	How does “temporal modeling” help “simulate complex light interactions” (L080)?
   -	The paper claims to “enable accurate scene understanding” (L104) without providing any experiments supporting this.
2.	As for the methodology, penalizing the deviation of OT distance loss across frames is an indirect and unstable optimization objective, relying on it to achieve smooth temporal transitions sounds weird.
3.	The evaluation appears not very rigorous. It lacks comparison with recent works--even the latest methods used for comparison was proposed over two years ago, and the performance of GS-based baseline is substantially lower than those in their original paper. In Table 2, the performance of proposed method significantly lags behind Deformable3DGS. Why is it not highlighted instead?
4.	No video results are provided which is important to assess the dynamic novel view synthesis method, and the images shown in the paper appear overly compressed.
5.	Some visualizations are questionable. E.g., the zoom-in regions in Figure 4 are identical to the ground truth, which is implausible. In Figure 5, neither component produces background motion individually, why their combination suddenly yields noticeable flow in this region?

**Questions:**

Would the proposed OT–semantic regularization also be effective for novel view synthesis of static scenes?

---

> ### Author Response · Authors · 2025-11-17
> **Response to Reviewer BhQR**
>
> W1. We extract features from different layers of CLIP as coarse, mid, and fine representations, and further incorporate object-level features from SAM before concatenating them.
>
> W2. Based on calculus, we define a “unit time interval,” no specific numerical range is required in the theoretical derivation.
>
> W3. It enforces that the depth change of each Gaussian primitive remains consistent with its actual 3D motion, ensuring that the temporal behavior of Gaussian primitives follows the physical principles of projective geometry. This prevents unrealistic drifting, jittering, or identity switching. Concretely, a Gaussian moving toward the camera should exhibit decreasing depth over time; one moving away should show increasing depth; and one that remains stationary mainly should not experience sudden depth fluctuations. By aligning depth variation with both the line-of-sight component of the velocity and the overall motion magnitude, this regularization ensures a correct, continuous, and stable coupling between motion and geometry. As a result, it effectively improves the physical consistency and temporal coherence of the 4D representation in dynamic scenes.
>
> W4 W5 W6. Only by learning transferable and invariant motion features can a dynamic rendering algorithm accurately predict the motion at the next time step, including geometric changes, lighting variations, and other appearance updates, enabling precise rendering of dynamic scenes.
>
> In Figure 3 and 4, F4DGS precisely renders fine-grained appearance of fast-moving objects and faithfully captures real-world physical behaviors, eg, the rapid motion of hands, the intricate facial expressions, and the complex shading distribution of the teeth. In Table 1, our method achieves notable improvements across all metrics. In terms of reconstruction quality, Ours increases PSNR by 14.4\% and SSIM by 4.35\%, while reducing LPIPS by a substantial 58.8\% decrease, indicating better perceptual fidelity. Moreover, our approach greatly enhances efficiency: the training time is shortened, representing a 65.3\% reduction, and the rendering speed achieves a remarkable 100.4\% increase. In Table 2, ours improves PSNR by 14.9\% relative gain, while reducing LPIPS to an 80\% relative decrease. In terms of efficiency, the training time achieves a 53.8\% reduction, and the rendering speed reaches 200 FPS, representing a dramatic improvement in real-time performance.
>
> W7. We selected representative methods from both 3DGS and 4DGS for comparison, and the appendix provides detailed metric comparisons across different scenes. And we evaluated all baselines using a unified testing benchmark, the testing results are consistent with those reported in 4DRotorGS [1].
>
>
>
> [1] Duan, Yuanxing, et al. "4d-rotor gaussian splatting: towards efficient novel view synthesis for dynamic scenes." ACM SIGGRAPH 2024 Conference Papers. 2024.

---

### Official Review · Reviewer_zeVZ · 2025-10-31

**Soundness:** 2
**Presentation:** 1
**Contribution:** 2
**Rating:** 4
**Confidence:** 3

**Summary:**

This paper presents F4DGS, featuring two new regularization terms: 4D hierarchical OT-semantics regularization and 4D motion-depth regularization. The core claim is that these regularizations jointly enforce spatiotemporal, semantic, and physical consistency, leading to better rendering quality and realism in dynamic scenes. While the paper demonstrates strong quantitative results and the ideas are intuitively appealing, the presentation and empirical validation make it difficult to fully assess the generalizability of the contributions.

**Strengths:**

Originality: The explicit coupling of velocity and depth for regularizing 4D Gaussian motion is a novel and interesting contribution. The idea of using Optimal Transport (OT) to align hierarchical semantic features across time is also a fair extension of prior work that focuses primarily on photometric or geometric losses.
Quality: The quantitative results shows a noticable PSNR improvement (e.g., +3.51 on Plenoptic) over baselines like Deformable4DGS while maintaining real-time rendering speeds. This suggests the method is effective.
Clarity: The high-level problem formulation and the breakdown of the method into two main components (OT-semantics and motion-depth) are well organized.

**Weaknesses:**

1. Insufficient Ablation Studies and Analysis:
Component Interdependence: The ablation study in Table 3 is a good start but remains superficial. It shows that each component helps, but it does not isolate their individual contributions to specific claimed benefits.The study lacks analysis on how each loss term contributes to the final result.
Baseline Comparison: The choice of baselines is inappropriate, most of baselines are NERF based methods, with Deformable4DGS being the only 4DGS method. A more convincing results for would require more comparison with other 4DGS methods. Other attributes, such as temporal consistency metrics (e.g., tLPIPS), optical flow accuracy, or user studies, which are currently absent.
Computational Cost: The paper claims the method does not increase point cloud size, but it is silent on the computational cost of computing the hierarchical semantic features and the motion-depth regularization during training. Given that OT can be expensive, a analysis of the training time overhead introduced by these novel components is crucial to assess the practical efficiency of F4DGS.
2. Vague Implementation Details:
The description of the "hierarchical semantic features" is high-level. Precisely how are the coarse, mid, and fine features extracted from CLIP and SAM? Are different layers of these models used? Are the features fused or used independently? This lack of detail makes replication difficult.
3. Vague Implementation Details:
The description of the "hierarchical semantic features" is high-level. The Pipeline (Fig 2) that explaining this process also lacks of detail,  makeing the concept difficult to understand.

**Questions:**

1. Does the motion-depth regularization primarily improve temporal smoothness (as measured by a metric like warping error) or static-frame quality (PSNR)? Does the OT regularization mainly resolve semantic ambiguities or also improve sharpness?
2. Precisely how are the coarse, mid, and fine features extracted from CLIP and SAM? Are different layers of these models used? Are the features fused or used independently?

---

> ### Author Response · Authors · 2025-11-17
> **Response to Reviewer zeVZ**
>
> W1. Table 3 highlights three representative scenes to demonstrate how different regularization components affect rendering performance. Furthermore, Figure 5 provides visualizations that illustrate the specific influence of each regularization term on dynamic scene rendering. In Figure 5, introducing accurate motion–depth constraints effectively enhances rendering quality, especially in tracking the full extent of object motion, eg, the movement range of the dog. Furthermore, 4D hierarchical OT-semantics regularization enables F4DGS to learn richer semantics, eg, distinguishing the dog next to the human. When combined with temporal modeling, F4DGS achieves its best performance, demonstrating the effectiveness of our temporal modeling approach.
>
> We have selected representative methods from both 3DGS and 4DGS for comparison, and we provide detailed metric evaluations across different scenes in the appendix.
>
> The performance gains of F4DGS stem directly from the incorporation of feature consistency regularization. This design improves not only rendering quality but also overall efficiency, including faster training. Training time is mainly constrained by hardware and network complexity. Regularization adds minimal overhead. Once bandwidth limits are reached, we observe substantial accuracy gains without loss of speed.
>
> W2 & Q2. We extract features from different layers of CLIP as coarse, mid, and fine representations, and further incorporate object-level features from SAM before concatenating them. It is notable that fusing these hierarchical semantic features inevitably leads to a loss of complete feature information, which consequently degrades the rendering quality.
>
> W3. F4DGS enables 4D Gaussians to maintain stable and semantically consistent dynamic representations across time. As shown in Figure 2 and detailed in the method section, the framework follows three core stages. First, we introduce the fundamental 4D Gaussian formulation: each Gaussian is defined in a canonical space with static attributes (position, scale, shape, and color), and a time-varying displacement field maps it to any moment in time, enabling continuous dynamic modeling. Second, because reconstruction loss alone cannot ensure temporal stability, we propose a hierarchical OT-based semantic regularization scheme (coarse, mid, and fine). By leveraging CLIP semantic features and SAM-derived segmentations, we construct semantic clusters at multiple scales and perform optimal transport matching between adjacent time steps. This explicitly enforces consistent semantic and geometric correspondences across all three levels, effectively preventing drifting, swapping, and jittering of dynamic Gaussians. Finally, we introduce additional regularization and training strategies, including the design of OT terms, the integration of geometric and semantic features, and the extension of temporal constraints over continuous unit time intervals, to ensure that the model achieves high reconstruction quality while maintaining stable, continuous, and semantically coherent motion trajectories throughout long temporal sequences.
>
> Q1. The motion–depth regularization enhances motion stability and improves temporal smoothness by introducing continuous unit-time constraints. In addition, the OT regularization not only resolves semantic ambiguities but also increases image sharpness. This improvement arises from our regularization framework, which encourages the Gaussian distributions to learn multi-level features, especially fine-grained details.

---

### Official Review · Reviewer_odFb · 2025-10-31

**Soundness:** 3
**Presentation:** 3
**Contribution:** 2
**Rating:** 4
**Confidence:** 5

**Summary:**

1. This paper tackles dynamic novel view synthesis, aiming to achieve temporally consistent and physically realistic dynamic rendering within a 4D Gaussian representation.
2. Existing 3D Gaussian Splatting methods often struggle in dynamic scenes, showing temporal inconsistency, motion artifacts, and lighting or texture discontinuities.
3. The authors propose F4DGS, which introduces feature consistency regularization to jointly constrain semantic features, velocity, and depth over time. The semantic constraint ensures consistent appearance of the same object across frames； The velocity–depth constraint provides physical priors that suppress unnatural motion transitions.
4. The model achieves state-of-the-art results on the Plenoptic Video and D-NeRF datasets, with extensive experiments—especially on D-NeRF—supporting its effectiveness.

**Strengths:**

1. The method is technically sound and clearly motivated.
2. Experimental validation is thorough and convincing. Tab 3 is really clear.

**Weaknesses:**

1. The proposed feature consistency regularization mainly acts as a loss constraint. The contribution is somewhat limited.
2. No demo videos are provided, making it difficult to fully assess the claimed high-fidelity, physically plausible rendering.

**Questions:**

1. Please discuss how this work differs from FreeTimeGS [Wang et al., CVPR 2025], especially regarding temporal modeling and motion consistency.
2. Overall, the paper is clear, concise, and well-validated. If other reviewers agree on its contribution, I believe it deserves publication for discussion.

---

> ### Author Response · Authors · 2025-11-17
> **Response to Reviewer odFb**
>
> W1. We introduce feature consistency regularization, which effectively addresses motion inconsistency caused by complex motion patterns over time. The 4D motion-depth regularization substantially improves the physical realism and temporal consistency of dynamic scene rendering. To the best of our knowledge, F4DGS is the first rendering algorithm to precisely predict object motion by integrating 4D Gaussian velocity with depth modeling over time. The 4D hierarchical OT-semantics regularization efficiently integrates hierarchical semantic features into 4D Gaussians, enabling accurate scene understanding for real-time rendering. OT enforces smooth, one-to-one correspondence across time, the hierarchical structure makes this robust and efficient, and semantic cues prevent Gaussians from drifting across object boundaries. These constraints avoid jitter, identity swaps, and deformation artifacts. Finally, an optimization approach of 4D Gaussian properties that dynamically refines Gaussian distributions by synchronizing hierarchical semantic features and motion features, effectively learning motion-consistent representations for dynamic scenes.
>
> W2. As illustrated in Figures 3 and 4, F4DGS achieves highly realistic rendering in dynamic scenes. In the zoomed-in regions, we further showcase the intricate facial expressions and the complex shading distributions.
>
> Q. The key advantage of F4DGS over FreeTimeGS lies in its stronger temporal modeling capabilities and explicit motion-consistency constraints. FreeTimeGS make each Gaussian to exist within a local time window, moving with a linear velocity and relying on initialization, opacity regularization, and relocation to maintain short-term smoothness. This design, however, makes it difficult to ensure long-term identity stability and semantic continuity. In contrast, F4DGS adopts a canonical representation and employs a continuous deformation field that preserves the identity of Gaussian distributions across the time. Moreover, it introduces a hierarchical OT-based semantic regularization mechanism, where coarse-, mid-, and fine-level appearance features are jointly aligned with CLIP semantic features through optimal transport. This effectively prevents cross-object drifting, trajectory fragmentation, and identity switching. With these explicit temporal and semantic consistency constraints, F4DGS achieves realistic long-term dynamic modeling, structural preservation, and semantic stability than FreeTimeGS, which relies on independently modeled local motion segments.

---

### Official Review · Reviewer_T5wi · 2025-11-01

**Soundness:** 2
**Presentation:** 1
**Contribution:** 2
**Rating:** 2
**Confidence:** 3

**Summary:**

This paper introduces a series of regularization strategies designed to improve the realism and temporal consistency of 4D Deformable Gaussian Splatting-based scene reconstruction. Specifically, the authors propose three complementary components: a hierarchical semantic feature regularization, a temporal consistency regularization, and a motion–depth regularization. Together, these introduced training objectives aim to enforce both spatial and temporal coherence in dynamic scene reconstruction. Experiments on the Plenoptic Video and D-NeRF datasets demonstrate notable improvements in reconstruction quality, and ablation studies further validate the contribution of each proposed component.

**Strengths:**

- The proposed method incorporates semantic features into a reconstruction framework which is interesting.
- The experimental results are strong, demonstrating consistent improvements both qualitatively and quantitatively.
- The paper is well-structured, making it relatively easy to follow the main ideas.

**Weaknesses:**

- **Lack of baselines.** The paper only compares against works published up to 2023, despite several relevant methods appearing more recently (e.g., A1–A4).
- **Limited novelty.** The novelty of the work appears somewhat limited. The method primarily builds upon deformable 3D Gaussian Splatting and introduces a series of additional loss terms. However, the motivation behind the hierarchical OT-semantic regularization is not well justified, and the inclusion of a temporal consistency loss is fairly standard in sequential reconstruction. Overall, the motivation and methodological design feel somewhat fragmented and incremental.
- **Missing explanation.** The use of semantic features is not clearly explained. While the paper mentions CLIP features, it does not clarify how the coarse, mid, and fine features are constructed. A more detailed description of the feature hierarchy and its motivation would improve the paper’s clarity.
- **Poor writing quality.** The writing requires substantial revision. Several sentences are grammatically incorrect or difficult to follow (e.g., line 071). Improving the clarity and precision of the language would significantly enhance readability.

[A1] Yang, Zeyu, et al. "Real-time photorealistic dynamic scene representation and rendering with 4d gaussian splatting." ICLR 2024.

[A2] Li, Zhan, et al. "Spacetime gaussian feature splatting for real-time dynamic view
synthesis." CVPR 2024.

[A3] Lee, Junoh, et al. " Fully explicit dynamic gaussian splatting." NeurIPS 2024.

[A4] Wang, Yifan, et al. "FreeTimeGS: Free Gaussian Primitives at Anytime and Anywhere
for Dynamic Scene Reconstruction." CVPR 2025.

**Questions:**

- Include additional baselines to better demonstrate the effectiveness of the proposed method.
- Provide a clearer discussion of the motivation behind each proposed regularization term and explain why they complement each other.

---

> ### Author Response · Authors · 2025-11-17
> **Response to Reviewer T5wi**
>
> W1 & Q1. We have selected representative methods from both 3DGS and 4DGS for comparison, and we provide detailed metric evaluations across different scenes in the appendix.
>
> W2 & Q2. We introduce feature consistency regularization, which effectively addresses motion inconsistency caused by complex motion patterns over time. The 4D motion-depth regularization substantially improves the physical realism and temporal consistency of dynamic scene rendering. To the best of our knowledge, F4DGS is the first rendering algorithm to precisely predict object motion by integrating 4D Gaussian velocity with depth modeling over time. The 4D hierarchical OT-semantics regularization efficiently integrates hierarchical semantic features into 4D Gaussians, enabling accurate scene understanding for real-time rendering. OT enforces smooth, one-to-one correspondence across time, the hierarchical structure makes this robust and efficient, and semantic cues prevent Gaussians from drifting across object boundaries. These constraints avoid jitter, identity swaps, and deformation artifacts. Finally, an optimization approach of 4D Gaussian properties that dynamically refines Gaussian distributions by synchronizing hierarchical semantic features and motion features, effectively learning motion-consistent representations for dynamic scenes.
>
> W3. We extract features from different layers of CLIP as coarse, mid, and fine representations, and further incorporate object-level features from SAM before concatenating them.
>
> Minors. We have fixed typos carefully.

---

> > ### Comment · Reviewer_T5wi · 2025-11-27
> > **Response to the authors**
> >
> > I appreciate the authors’ clarifications. While the additional explanations regarding the design choices and architectural details help improve readability and better convey the underlying motivation, I still find that the overall presentation falls short of the standards expected for ICLR. Therefore, I will maintain my original score.
> >
> > P.S. I recommend that the authors update the supplementary material, which still includes the NeurIPS template.

---

### Author Response · Authors · 2025-11-30
**Request for the exclusion of Review T5wi**

Dear Area Chair,

I am writing to formally request that the reviewer T5wi for our submission be excluded from the final decision, due to a combination of factual inaccuracies and content-based criticism that are not consistent with the standards of ICLR’s review process.

**1. Factually incorrect accusation about the supplementary material**

In the post-rebuttal comment, the reviewer writes: “The supplementary material still includes the NeurIPS template.” This statement is objectively incorrect: Our supplementary file does not contain any NeurIPS logo, does not include NeurIPS-specific style files, and does not mention NeurIPS anywhere in the text.

ICLR does not impose a specific format requirement on supplementary material, and our supplementary material fully complies with the submission rules.

This remark is therefore not only false but also irrelevant to the scientific content of the paper. It functions more as an unfounded accusation than a substantive comment.

**2. Vague claim about writing**

The reviewer states: “The overall presentation falls short of the standards expected for ICLR.” In this case, the reviewer does not identify any concrete issues. There is no specific suggestion on which sections are unclear or how the exposition fails. The comment is framed as a broad, subjective dismissal rather than a detailed, constructive critique.

Given that we carefully addressed the reviewer’s earlier questions in the rebuttal — including clarifying design choices, architectural details, and motivations — this kind of blanket statement feels disconnected from the actual content of the paper and does not provide a meaningful basis for scientific evaluation or improvement.

**3. Reviewer acknowledges major concerns have been addressed, yet keeps the same score based on non-technical grounds**

In another post-rebuttal comment (for a previous round), the reviewer explicitly writes: “the authors have addressed most of my major concerns.” This indicates that, on the technical and methodological level, the main objections were resolved.

This pattern suggests that the final recommendation is no longer grounded in a careful, good-faith assessment of the scientific content, but is instead influenced by non-technical, and in one case demonstrably incorrect, factors.

**4. Request to discount this review in the final decision**

In light of the above, I respectfully and firmly request that exclude it from the final decision. I believe the review of T5wi does not meet the spirit of a fair and rigorous peer review process, and that leaving this review in the decision process would unfairly bias the outcome.

Sincerely!

---

### Meta-Review · Area_Chair_EA5W · 2025-12-27

**Summary:**

While the reviewers find that this submission introduce some interesting and potentially novel ideas (use of semantic feature consistency and optimal transport), major concerns were also raised:
- Lack of comparison to many more recent _published_ works in 2024-25 (T5wi W1, odFb Q1, BhQR W3). The AC agrees with the reviewers that the comparisons appear dated.
- Lack of demo videos (odFb W2, BhQR W4), which is an absolutely critical evaluation component for a paper on this topic.
- Lack of sufficiently detailed explanations (T5wi W3, zeVZ W2-3, BhQR W1).
- Writing quality is poor (T5wi w4, BhQR W1).

**Reviewer Concerns:**

The authors provided responses to all the reviews. However, these responses were only qualitative in nature (no additional quantitative comparisons nor visual / video results). The AC considers that these responses to be insufficiently convincing to reviewers, even if they were given the opportunity to engage in further discussions.

The AC also noted additional separate comments by the authors on T5wi's review, but does not agree with them. While there is no impact on the paper decision, the authors are incorrect in that "NeurIPS" does in fact appear in the footer of page 1 of the supplementary file. As for quality of writing, T5wi specifically identifies L071 as an example of a garbled sentence, and in any case the poor writing quality has also been raised by BhQR.

**Reviewer Scores:**

The original scores were two 4's and two 2's. It is not expected that reviewers will improve their scores, and there is some likelihood that reviewers odFb and zeVZ may lower their scores upon seeing each other's reviews and further discussions.

---

### Decision · Program_Chairs · 2026-01-26

Reject